# Aging Disrupts Circadian Rhythms in Mouse Liver Mitochondria

**DOI:** 10.3390/molecules28114432

**Published:** 2023-05-30

**Authors:** Wei Xu, Xiaodong Li

**Affiliations:** College of Life Sciences, Wuhan University, Wuhan 430072, China

**Keywords:** circadian rhythms, peripheral clock, inflammatory response, aging, senescence

## Abstract

The circadian clock regulates daily changes in behavioral, endocrine, and metabolic activities in mammals. Circadian rhythms in cellular physiology are significantly affected by aging. In particular, we previously found that aging has a profound impact on daily rhythms in mitochondrial functions in mouse liver, leading to increased oxidative stress. This is not due to molecular clock malfunctions in peripheral tissues in old mice, however, as robust clock oscillations are observed therein. Nonetheless, aging induces changes in gene expression levels and rhythms in peripheral and probably central tissues. In this article, we review recent findings on the roles of the circadian clock and the aging process in regulating mitochondrial rhythms and redox homeostasis. Chronic sterile inflammation is implicated in mitochondrial dysfunction and increased oxidative stress during aging. In particular, upregulation of the NADase CD38 by inflammation during aging contributes to mitochondrial dysregulation.

## 1. Introduction

Intrinsic circadian oscillations are present in certain nuclei of the central nervous system (e.g., the suprachiasmatic nucleus (SCN)) and most peripheral tissues in mammals [1]. At the molecular level, circadian oscillations are driven by core clock proteins that serve as transcription factors (TFs). In particular, the clock proteins CLOCK and BMAL1 dimerize to transcriptionally activate other core clock genes, such as *Per1/2* and *Cry1/2*, whose protein products antagonize the actions of CLOCK/BMAL1, forming a negative feedback loop [1]. CLOCK/BMAL1 also activates the transcription of clock genes *Nr1d1/2*, whose protein products inhibit *Bmal1* transcription, forming another feedback loop [1]. In addition to regulating their own transcription to sustain circadian oscillation, clock proteins also exert broad control over other genes. As a result, about 10% of genes expressed in peripheral tissues are rhythmic, and the rhythmic genes are involved in nearly all aspects of cellular functions (e.g., metabolism, immune defense, and cell cycle regulation) [2,3]. Clock functions are tissue-specific, and gene expression rhythms vary greatly across tissues. Indeed, the genomic binding of clock proteins is tissue-specific [4,5]. Those sites are found within open chromatin regions established by tissue-specific TFs (e.g., pioneer TFs) and some ubiquitously expressed TFs [6].

Core clock genes typically harbor multiple cis-elements for clock proteins themselves, a strong control thought to ensure robust and resilient clock oscillation across various cell types [6]. However, circadian rhythms are not regulated by the intrinsic tissue clock alone. Extrinsic cues, including those derived from the SCN (neural, humoral, and behavioral cues such as body temperature) and communicating signals from other tissues, also regulate gene expression rhythms [7]. Previous studies emphasize the influence of extrinsic cues on intrinsic clocks per se. Indeed, extrinsic cues can engage TFs (e.g., CREB and SRF) that regulate the transcription of the clock genes [1]. Given that TFs typically have many genomic binding sites, they could also influence other genes besides clock genes. Indeed, in the absence of rhythmic cues from the SCN and other tissues, the intrinsic clock can only sustain the rhythms of a limited set of genes (e.g., core clock genes and some clock-controlled genes). Those results indicate that clock proteins often collaborate with other TFs in rhythm regulation [6]. The collaboration can occur within the same enhancer co-bound by clock proteins and other TFs; it can also be achieved through the looping of distinct enhancers bound by clock proteins and other TFs, respectively, to promoters of their common target genes. Such combinatorial control permits plasticity in circadian rhythm regulation by allowing TFs other than clock proteins to control rhythms [6]. For example, a high-fat diet reprograms liver gene expression via PPARγ and SREBP but leaves the liver clock intact [8,9]. Lung adenocarcinoma in young mice releases cytokines to reprogram liver gene expression without disturbing the liver clock [10]. Interestingly, aging also leads to extensive reprogramming of daily gene expression in peripheral tissues of mice and man, while clock oscillations remain normal therein [11,12,13,14].

Mitochondria are the hub of metabolism and a significant source of reactive oxygen species (ROS). Their dysfunction is a hallmark of aging [15]. Previously, we found robust daily rhythms of mitochondrial functions in the livers of young mice, which are regulated by the clock [16]. Moreover, we found that mitochondrial rhythms are disrupted in the livers of old mice while normal clock oscillation persists [16]. Those results indicate that mitochondrial functions are regulated not only by the clock but also by age-related mechanisms. In particular, aging disrupts mitochondrial redox homeostasis and elevates oxidative stress. In this article, we first provided an overview of ROS production and antioxidant defense. We then focused on clock roles in regulating mitochondrial functions, including redox homeostasis. Finally, we discussed mechanisms that dysregulate mitochondrial functions during aging and the consequences of age-related increases in oxidative stress to circadian timing functions.

## 2. ROS and Oxidative Stress

ROS is an umbrella term for free radicals (species with at least one free electron, e.g., the superoxide anion O_2_^•−^) and non-radical species derived from oxygen (e.g., H_2_O_2_) [17]. Electron leakage from protein redox centers can produce O_2_^•−^ [18]. Mitochondrial dehydrogenases and respiratory complexes of the electron transport chain (ETC) are all O_2_^•−^ production sites [19]. O_2_^•−^ can also be directly produced by enzymes such as NADPH oxidases (NOXs), cytochrome P450 enzymes (CYPs), and xanthine oxidases [18]. O_2_^•−^ can react with nitric oxide (NO) to form peroxynitrite (ONOO^−^), and oxidants derived from nitric oxide are called reactive nitrogen species (RNS). O_2_^•−^ can be dismutated to H_2_O_2_ spontaneously or by superoxide dismutase (SOD) enzymes. H_2_O_2_ is also enzymatically produced by oxidases. H_2_O_2_ can oxidize the sulfhydryl (–SH) group of cysteine in proteins into sulfenic acid (–SOH). The cysteine –SH group can also be converted into –SNO (S-nitrosylation) by NO. Both –SOH and –SNO can react with glutathione, either spontaneously or by enzymatic actions (S-glutathionylation); deglutathionylation is carried out by specialized enzymes [20]. S-nitrosylation and S-glutathionylation are thought to protect proteins from further oxidative and/or nitrosative modification. Indeed, the sulfenic acid group can be further oxidized by H_2_O_2_ into sulfinic acid (–SO_2_H) and sulfonic acid (–SO_3_H). While the former can be reversed by sulfiredoxin (SRX) [21], the latter form is not reversible. Importantly, H_2_O_2_ leads to the production of the hydroxyl radical –OH, a powerful oxidant that damages proteins, lipids, DNAs, and sugars. –OH can directly oxidize various protein residues. –OH and free radicals derived from fatty acids damage lipids through lipid peroxidation reactions [22], which can lead to ferroptosis [23]. The lipid peroxidation products (e.g., 4-HNE) are also highly reactive toward proteins and other biomolecules. Oxidization of proteins by –OH and lipid peroxidation products leads to protein carbonylations that are not readily reversed and can disturb cell functions.

ROS and RNS, as well as some forms of molecular damage caused by them, can play signaling roles (see below). However, excessive ROS/NOS and molecular damages can disturb redox signaling and control, resulting in oxidative stress [24]. Dysfunctional biomolecules need to be eliminated, a task carried out by the proteasome and via autophagy. Hyperoxidized, S-glutathionylated, and carbonylated proteins can be degraded by the 20S proteasome in an ATP- and ubiquitin-independent manner [25]. Some damaged proteins and organelles (e.g., mitochondria) are degraded through autophagy [25]. RNS plays complex roles in mitophagy [26]. We focus below on the biology of ROS.

## 3. Redox Homeostasis: Maintaining the Balance between ROS Production and Antioxidant Defense

In addition to eliminating oxidative damage via proteasome and autophagy, cells are endowed with antioxidant defense systems to reduce ROS production and oxidative stress. They include the glutathione (GSH), glutaredoxin (GRX), and thioredoxin (TRX) systems. While all systems use NADPH as the ultimate reducing power, each can have specialized functions [24]. For example, peroxinredoxin 6 (PRDX6), glutathione S-transferases (GSTs), and glutathione peroxidase 4 (GPX4) can restrain damages by lipid peroxidation. GSTs and GRXs are mainly responsible for protein S-glutathionylation and its reversal, respectively. GPXs and PRDXs mitigate oxidative stress by catalyzing H_2_O_2_ removal. In particular, PRDX3 plays a prominent role in eliminating H_2_O_2_ in mitochondria. A cysteine residue is found at the catalytic center of PRDXs, and its sulfhydryl group is oxidized by H_2_O_2_ to sulfenic acid. Upon oxidization of typical 2-Cys PRDXs (PRDX1-4), intermolecular disulfide bonds are formed, which are resolved by TRX [27,28]. The dimer/monomer ratio of PRDX can be used as an indicator of oxidative stress. Sulfenic acid at PRDX active site can be further oxidized by H_2_O_2_ into sulfinic acid (antagonized by SRX) and even sulfonic acid. However, under physiological conditions, hyperoxidized PRDXs are typically of low abundance in cells and tissues, and they can be degraded by the 20S proteasome [29,30].

## 4. ROS and Oxidative Stress Play Signaling Roles

ROS play signaling roles in physiology [17,31]. For example, O_2_^•−^ can release Fe^2+^ from Fe-S (iron-sulfur) clusters in proteins to affect metabolism [31]. ROS can also activate uncoupling proteins (UCPs) in the mitochondrial inner membrane to dissipate the H^+^ electrochemical gradient for heat production instead of ATP synthesis [32,33]. ROS production by the ETC is associated with mitochondrial energetics and varies by ATP supply and demand [34,35]. A high electrochemical potential of H^+^ (the protonmotive force, pmf) promotes ROS production by the ETC [35]. The buildup of pmf is favored under nutrient-rich conditions (e.g., using glucose or pyruvate as substrate) due to abundant NADH and FADH2 supply to the ETC. Increased ROS, in synergy with other factors such as mitochondrial [Ca^2+^] elevation, can trigger the transient opening of the mitochondrial permeability transition pore (mPTP) [36] to decrease pmf and halt ATP production [34]. Thus, ROS plays a signaling role in maintaining ATP homeostasis [34]. The molecular identity of mPTP and regulation of its opening by ROS, Ca^2+^, and H^+^ has been elucidated [34,37,38,39]. Interestingly, transient mPTP opening triggers a burst of O_2_^•−^ production, known as “ROS-induced ROS release” [40] or “mitoflash” [41]. Mitoflash frequency signals basal ROS and oxidative stress levels.

In addition to its role in metabolism, ROS also regulates signal transduction. This role is carried out mainly by H_2_O_2_, which can oxidize redox-sensitive cysteines in target proteins. For example, AMPK, the metabolic regulator activated by glucose shortage and low energy charge [42], is subject to redox-based regulation. H_2_O_2_ can oxidize cysteines in AMPKα to hinder its activation by upstream kinases [43]. PTEN, which dephosphorylates phosphatidylinositol 3,4,5-triphosphate (PIP3) to inhibit PI3K signaling, is also subject to redox-based regulation. Like AMPK, PTEN activity is inhibited when H_2_O_2_ oxidizes its active site cysteine, turning the sulfhydryl group (–SH) into sulfenic acid (–SOH). The –SOH group reacts with the –SH group of another cysteine in PTEN to form an intramolecular disulfide bond, thus inactivating PTEN [44]. Protein tyrosine kinases (PTKs) are activated by cytokines, growth factors (e.g., insulin), and T- and B-cell receptor stimulation; protein tyrosine phosphatases (PTPs) negatively regulate PTK signaling [45]. During PTK activation, O_2_^•−^ is produced by NOXs and converted to H_2_O_2_; PTKs phosphorylate and inactivate PRDXs to allow localized H_2_O_2_ accumulation [46,47]. H_2_O_2_ oxidizes the active site cysteine to inactivate PTP, thus prolonging PTK signaling. PTP oxidization leads to the intramolecular formation of a disulfide (e.g., SHP2) or a sulfenyl–amide bond (e.g., PTP1B). Oxidation and inactivation of AMPK, PTEN, and PTPs by H_2_O_2_ are transient, and they can be reversed by reducing agents such as TRX and GSH [48].

## 5. ROS and Oxidative Stress Regulate Gene Transcription

In addition to modulating signal transduction, ROS and oxidative stress also regulate transcription. A well-known example is their activation of the KEAP1-NRF2 system for the antioxidant defense to maintain redox homeostasis. From a mechanistic point of view, antioxidant defense is related to xenobiotic detoxification, which consists of three phases [49]. Phase I enzymes (e.g., CYPs) can activate xenobiotics, and some products that are electrophilic are conjugated with reducing agents by phase II enzymes (e.g., GSTs); conjugation products are excreted through transporters encoded by phase III genes. Some phase I (e.g., ALDH2) and II (e.g., GST) enzymes can detoxify lipid peroxidation products. NRF2 is a TF that can upregulate phase II genes. Normally, KEAP1 complexes with NRF2 at a 2:1 ratio and promotes NRF2 ubiquitination and degradation by the proteasome. During xenobiotic detoxification, electrophilic products from phase I reactions can oxidize cysteine residues in KEAP1, leading to the formation of disulfide bonds between KEAP1 dimers [50]. The conformational changes in KEAP1 liberate NRF2. Evading degradation, NRF2 accumulates and moves into the nucleus to upregulate phase II genes [50]; NRF2 also regulates iron homeostasis to control ferroptosis [51]. CYPs are also ROS sources. Similar to electrophiles, H_2_O_2_ oxidizes cysteine residues in KEAP1 to enable NRF2 for transcription regulation [52]. Thus, the KEAP1-NRF2 system is employed as a negative feedback mechanism to defend against both electrophiles and ROS oxidants. NRF2 can also be activated when p62/SQSTM1 sequestrates KEAP1 for selective autophagy [53]. ROS and oxidative stress also regulate other TFs, such as HIF-1α, for adaptions to hypoxia [54]. ROS also facilitates inflammatory responses and NF-kB activation [17]. As discussed later, inflammatory TFs, including NF-kB, are most probably involved in age-related reprogramming of daily gene expression.

## 6. Reciprocal Regulation between the Circadian Clock and Redox Homeostasis

Given the broad control over cellular functions by the circadian clock, it is not a surprise that daily changes in redox regulation are observed in various tissues [55]. The liver plays an essential role in metabolism and is an important model of peripheral clocks. Microarray studies revealed that some rhythmic genes in mouse liver are involved in xenobiotic detoxification [56,57]. Those include phase I (e.g., CYPs), II (e.g., *GST*), and III (e.g., ABC transporters) genes. In addition, xenobiotic receptors (e.g., *Car*) and *Alas1* (for the biosynthesis of heme, the prosthetic group of CYPs) are also rhythmically expressed. The circadian clock also regulates antioxidant defense genes *Aldh2* and *Nqo1* [58]. The pentose phosphate pathway (PPP) is the major cellular source of NADPH. The circadian clock indirectly regulates PPP genes and NADPH production via PPARδ [59,60]. The clock also regulates NRF2 to control GSH-mediated antioxidant defense [61,62]. Finally, the clock regulates autophagy [63,64], which is known to reduce oxidative stress. The ULK1/2 kinases not only promote autophagy (peaking during late daytime in the mouse liver) but also promote NADPH production by the PPP [65]. Those results highlight the critical roles of the circadian clock in coordinating redox regulation across the day.

In a reciprocal manner, redox states also affect the clock. For example, the redox states of NAD(H) and NADP(H) influence clock proteins’ DNA binding [66], and perturbing the PPP can alter clock dynamics [67]. NRF2 can regulate clock genes *Cry2* and *Nr1d1* [67,68]. Another link between redox states and the circadian clock is the redox regulation of ion channel functions in the SCN to influence its neural output [69].

Reciprocal regulation between the clock and redox states most probably has adaptive values. For most animals, a feeding/fasting cycle co-varies with their sleep/wake cycle across the day. Many liver functions (e.g., bile production) are related to feeding and exhibit robust daily changes. Clock-controlled induction of xenobiotic detoxification genes in anticipation of feeding could be beneficial for organismal health [57]. During the glucose-rich feeding phase, increased ROS production may facilitate insulin signaling to regulate liver metabolism. On the other hand, during the fasting phase, glucose shortage can activate AMPK to promote autophagy and fatty acid oxidation (FAO) as an alternative energy source, a metabolic adaptation that favors oxidative stress reduction (see next section). Thus, circadian redox homeostasis is manifested as a daily rhythm of redox states, which is an integral part of daily changes in metabolism and physiology. Clock gene mutations significantly disturb redox homeostasis. In particular, deficiencies within the positive limb of the clock (i.e., *Clock*, *Bmal1*, and *Npas2*) decrease the expression of antioxidant defense genes and increase oxidative stress in young mice [16,58,61,70]. A premature aging phenotype is observed in *Bmal1* knockout mice [71,72].

Intriguingly, metabolic and redox oscillations are found in red blood cells devoid of nuclear transcription and cells lacking functional molecular clocks [29,73]. Nonetheless, an intact clock sustains more robust and widespread circadian oscillations at molecular and cellular levels [74].

## 7. Mitochondrial Functions Are Rhythmic and Regulated by the Circadian Clock in Young Mice

Except for the 13 OXPHOS subunits encoded in mtDNA, all other mitochondrial proteins, including those involved in mtDNA replication, transcription, and mitochondrial protein translation, are encoded by the nuclear genome [75]. Some of those genes appear under clock control [76,77]. We and others found daily changes in the expression of OXPHOS genes encoded not only by the nuclear genome but also by mtDNA in the livers of young mice [16,78]. The daily changes in OXPHOS protein composition and other mitochondrial rhythms (see below) are probably optimized for daily changes in nutrient supply and energy demand associated with the feeding-fasting cycle. Depending on the feeding state, mitochondria switch fuel choice between pyruvate and fatty acid [79]. That mitochondrial energetics and fuel usage vary over the day is clearly evidenced by daily changes in oxygen consumption rate and the respiratory exchange ratio in mice and men [78,80,81]. Mitochondrial respiratory activities in vitro also change across the day, but the peak phases differ by substrates, consistent with daily changes in fuel usage [78,82]. Some mechanisms of mitochondrial fuel selection are known. For example, reversible phosphorylation of pyruvate dehydrogenase (PDH) controls mitochondrial pyruvate use. PDH is inactivated upon phosphorylation by PDH kinases and re-activated by a Ca^2+^-activated phosphatase [83]. Ca^2+^ influx promotes pyruvate oxidation [83] and is limited by MICU1 (regulating Ca^2+^ influx via MCU) and OPA1 present at the cristae junction [84]. We found that phosphorylation of PDH-E1α in the livers of young mice is increased during daytime, indicating that mitochondrial pyruvate oxidation is reduced during fasting [16]. Mitochondria undergo structural changes in response to nutrient availability [85]. Under nutrient-poor situations, OPA1 promotes the fusion of the mitochondrial inner membrane and intracristal assembly of OXPHOS complexes and supercomplexes to facilitate ATP production [86,87]. We found daily changes in mitochondrial OPA1 abundance in mouse liver, with higher levels at daytime [16], consistent with the role of OPA1 in promoting FAO [88].

OPA1 not only promotes FAO but also curtails oxidative stress [89]. ROS production rate by mitochondria in vitro is low during FAO [90]. We found that mitochondrial oxidative stress in vivo, judged by the degree of PRDX3 dimerization in young mouse liver, is decreased during daytime, with a nadir at ZT10 [16]. From a metabolic point of view, mitochondrial dehydrogenases and respiratory complexes are differentially employed when different respiratory substrates are used, so the ROS production rate would vary by substrate, as indeed observed in vitro [90]. Low ROS production rate during FAO could be accounted for by several mechanisms. First of all, FAO produces NADH and FADH2 at an equal molar ratio. Compared to NADH, FADH2 derived from FAO donates its electrons (via ETF) to the ETC downstream of ROS production sites in complex I, thus favoring less ROS production [91]. Other mechanisms exist to restrain oxidative stress during FAO. For example, during FAO in the liver, TCA cycle metabolites are depleted by gluconeogenesis, and acetyl-CoAs from FAO are diverted to ketogenesis. As a result, less FADH2, NADH, and ROS are made by the TCA cycle in the liver during FAO. Moreover, a limited NADH supply to the ETC during FAO would raise the NAD^+^/NADH ratio, thus activating sirtuins [92]. Notably, the mitochondrial sirtuin SIRT3 plays a role in reducing oxidative stress. SIRT3 deacetylates various proteins [91], such as ETC proteins, FAO, and antioxidant defense enzymes (e.g., IDH2 and SOD2). SIRT3 increases the efficiency of ETC electron transfer to reduce ROS production and also promotes antioxidant defense [91]. We found that acetylation levels of SIRT3 targets change over the day, reaching the lowest levels near ZT10, concurrent with the nadir of mitochondrial oxidative stress in the livers of young mice [16]. Overall, the daily rhythm in mitochondrial redox states is closely integrated with daily metabolic changes in mouse liver, such that low oxidative stress level is associated with FAO during fasting.

Consistent with clock control of mitochondrial functions [93], most mitochondrial rhythms we found in the livers of young mice, including the redox rhythm, are disrupted by the *Clock*^△19^ mutation [16]. The circadian clock also controls a redox rhythm at the whole cell level [94].

## 8. Mitochondrial Rhythms in Mouse Liver Are Disrupted by Aging despite Normal Circadian Clock Oscillation

Intriguingly, we found that the mitochondrial rhythms evident in young mice are disrupted by aging, due mainly to rhythm damping by age-related changes during the daytime, especially at ZT10 [16]. For example, mtDNA transcripts in the livers of old mice are much less abundant at ZT10, the peak time in young mice. Age-related decrease in mtDNA transcripts is also seen in mouse muscle, where nuclear-encoded OXPHOS transcripts are less affected by aging [95]. Changes in OXPHOS subunit composition probably lead to inefficient electron transfer and energy production and increased ROS production in various mouse tissues. Meanwhile, we found that SIRT3 target acetylation in the livers of old mice is increased at ZT10, the nadir of corresponding rhythms in young mice. Aging abolishes the mitochondrial redox rhythm, owing to a prominent increase of oxidative stress in the livers of old mice at ZT10, the nadir of PRDX3 dimerization in young mice. The age-related disruption of mitochondrial rhythms, however, is not associated with overt clock defects. We found that clock gene rhythms remain normal with age in mouse liver [16], consistent with other studies on mouse peripheral tissues [11,12,13] and the SCN [96,97,98]. Those results indicate that aging disrupts mitochondrial rhythms through molecular mechanism(s) downstream of the circadian clock. Indeed, deep sequencing studies revealed that, while robust clock oscillations are preserved in peripheral tissues, aging induces extensive changes in gene levels and rhythms therein [11,12,13]. Age-related global changes in SCN gene expression await future studies. Some genes related to mitochondrial functions in various tissues, including the few we identified in mouse liver [16], are expected to be affected by aging. Such genes remain to be fully characterized.

Maintaining proper circadian rhythms at the organismal and cellular levels is beneficial to health. For example, time-restricted feeding (tRF) enables robust daily rhythms to prevent metabolic diseases in young mice and men [99]. Age-related changes in gene expression clearly disturb metabolic processes, as evidenced by decreased rhythm amplitudes of many metabolic genes in the livers of old mice [13]. Such changes can be ameliorated by caloric restriction (CR), and the tRF factor contributes to the lifespan-extending effect of CR in mice [13]. The extent to which mitochondrial rhythms are affected by CR during aging remains to be determined.

## 9. Mechanism(s) for Age-Related Dysregulation of Mitochondrial Rhythms

The mechanisms for age-related reprogramming of daily gene expression are under current investigation (Figure 1). Gene expression is known to be controlled by both the circadian clock and other TFs, either independently or in combination [6]. Conceivably, some TFs whose activities are altered by aging take part in reprogramming daily gene expression. Plausible candidates are TFs (e.g., NF-kB, IRFs, and STATs) activated by chronic inflammation, a hallmark of aging [15]. Indeed, inflammatory response genes are enriched in the tissues of old mice [11,13]. Unlike the strong immune response that disrupts the molecular clock after an LPS challenge [100], age-related sterile inflammation is of low grade and thus may not significantly disturb the molecular clock [101], at least at the early stage of aging examined [11,12,13,16].

One notable gene target of inflammatory signaling is CD38, a NADase whose up-regulation contributes to the age-related decline in NAD^+^ levels in mouse tissues [102]. Low NAD^+^ levels lead to insufficient sirtuin activation that has pleotropic effects [103]. For example, SIRT6 restrains NF-kB signaling, and its deficiency leads to premature aging. SIRT1 deficiency promotes P53 activation and cellular senescence. SIRT1 also deacetylates PGC1α to activate the transcription of genes involved in mitochondrial functions [104]. Insufficient SIRT1 activation during aging may disturb mitochondrial rhythms. Decreased SIRT3 activation may be particularly relevant to the disruption of mitochondrial redox rhythms during aging, as evidenced by the increase in both SIRT3 target acetylation and mitochondrial oxidative stress in the livers of old mice [16]. SIRT3 also promotes mitobiogenesis via deacetylating TFAM, a protein essential for mtDNA maintenance and transcription [105]. Decreased SIRT3 activation probably contributes to the reduction of mtDNA transcripts during aging [16].

Increased oxidative stress is implicated in age-related dysregulation of SCN neural activity, which impairs circadian timing functions at the system level [106,107]. Age-related disturbance of redox homeostasis could also impact signaling pathways in various tissues. For example, AMPK and mTOR are alternatively activated across the day in young mice, in part due to the influence of the redox rhythm. Dysregulation of AMPK and mTOR during aging could induce changes in metabolism and gene expression [108]. Finally, ROS causes damage to DNA, which activates ATM to regulate the DNA damage response (DDR) of DNA repair, cell cycle arrest and/or apoptosis [109]. ATM is also activated by ROS and promotes NADPH production and autophagy to lower oxidative stress [109]. ATM activation in young animals is expected to be rhythmic across the day to maintain genome integrity. mtDNA is thought to be particularly vulnerable to damage due to its proximity to mitochondrial ROS. However, antioxidant defense, mitochondrial DNA repair, mitochondrial dynamics (fusion and fission), and mitophagy appear sufficient to maintain the integrity of mtDNA at least to the early stage of aging [95,110,111]. Nonetheless, during aging, loss of redox homeostasis and chronic increase in oxidative stress could favor persistent ATM activation, a condition under which ATM and other DDR proteins promote SASP (senescence-associated secretory phenotype) in cells to exacerbate systemic inflammation [112]. Such a vicious cycle may eventually lead to the accumulation of DNA damage and loss of proteostasis in various tissues, which can aggravate cellular dysfunctions and age-related diseases.

## 10. Future Perspectives

From the perspective of circadian rhythms, mitochondrial functions are regulated by both the clock and age-related mechanisms. In young animals, mitochondrial functions and redox states are rhythmic and integrated with daily changes in metabolic activities. Mitochondrial rhythms are disrupted during aging, when redox homeostasis is disturbed due to unbalanced ROS production and antioxidant defense, resulting in a net increase in oxidative stress. Age-related disruption of mitochondrial rhythms is associated with global changes in gene expression levels and rhythms in spite of intact tissue clocks. Such reprogramming of daily gene expression is a manifestation of plasticity in circadian rhythm control, and it involves TFs activated during aging (e.g., inflammatory TFs). How inflammatory TFs participate in reprogramming mitochondrial rhythms during aging should be characterized in detail in future studies. It is also of interest to examine the effect of CR on mitochondrial rhythms during aging. Finally, the phases of mitochondrial rhythms should be taken into account when administering daily pharmaceutical interventions aimed at healthy aging [108].

## Figures and Tables

**Figure 1 molecules-28-04432-f001:**
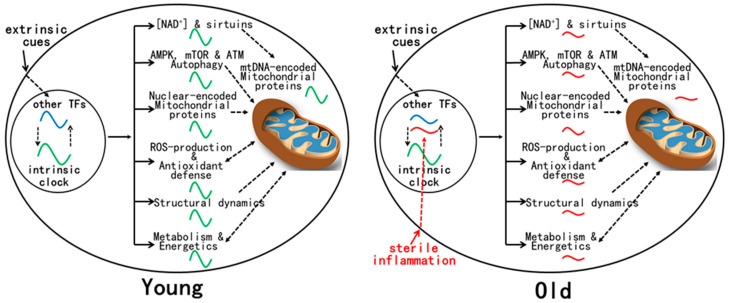
Mitochondrial functions are regulated by the circadian clock and other TFs in young animals. Such combinatorial regulation coordinates robust daily changes in mitochondrial protein composition, structural dynamics, energetics, and metabolism. There is a clear mitochondrial redox rhythm that is integrated with daily rhythms in metabolism. During aging, age-related activation of certain TFs, such as inflammatory TFs, leads to the reprogramming of daily gene expression that has profound effects on mitochondrial physiology, leading to the disruption of most mitochondrial rhythms that are evident in young animals. Such dysregulation of mitochondrial functions is associated with metabolic changes and disturbed redox homeostasis.

## Data Availability

Not applicable.

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
