# Peer review of "Aging Disrupts Circadian Rhythms in Mouse Liver Mitochondria"

_molecules, 2023, doi:10.3390/molecules28114432_

Round 1

Reviewer 1 Report

Dear Authors, 

Your MS 'Aging disrupts circadian rhythms in mouse liver mitochondria' is a substantial, well-written detailed review devoted to a very important effect of aging on circadian rhythms in mammals. The review contains a careful comparative analysis of the effect of oxidative stress on circadian rhythm and of the mechanisms of age related mechanisms of mitochondrial rhythms. The MS is very well logically structured. I recommend the following minor changes to be made before accepting the MS: 

The formula of superoxide should be written as O2·- everywhere in the text.

The formula for hydroxyl radical is typically ·OH

Ln 94: An explicit definition of oxidative stress is required here. The phrase "Molecular damage by ROS/RNS results in oxidative stress' is, generally speaking, incorrect. You should talk here about the imbalance between the oxidative and repair/antioxidant components of this phenomenon. 

In References, 2 and 7 should have page numbers. 

The quality of English Language is high, only minor changes are required:

Ln 34: should be 'pioneer'

Ln 37: 'cell types' sounds vague. Maybe: 'various cell types'

Ln 42: should be 'engage'.

Ln 49: should be 'can also'

Ln 53: format the spaces after 'PPPA'

Lns 132 and 133: the phrase 'The molecular identity... ' should be revised. Suggestion: "The molecular identity of mPTP and regulation of its opening by ... have been elucidated'.

Ln 160: 'produce... products' - this part should be revised

Lns 164 and 165 - should be 'NF-kB'

Ln 197 - should be 'influence' 

Ln 245 - should be 'curtails'

Ln 352: should be '... ; it involves...'

Ln 354 - should be 'enables'

Author Response

The formula of superoxide should be written as O2·- everywhere in the text.

Response: We revised our manuscript according to the suggestion.

The formula for hydroxyl radical is typically ·OH

Response: We revised our manuscript according to the suggestion.

Ln 94: An explicit definition of oxidative stress is required here. The phrase "Molecular damage by ROS/RNS results in oxidative stress' is, generally speaking, incorrect. You should talk here about the imbalance between the oxidative and repair/antioxidant components of this phenomenon. 

Response: We thank the reviewer for this important point. We revised our manuscript by stating “ROS and RNS as well as some forms of molecular damages caused by them can play signaling roles (see below). However, excessive ROS/NOS and molecular damages can disturb redox signaling and control, resulting in oxidative stress.” This definition of oxidative stress is proposed by Dr. DP Jones.

In References, 2 and 7 should have page numbers. 

Response: We could not obtain page numbers. We added DOI info instead. We also added a few new references to address issues raised by other reviewers.

Comments on the Quality of English Language

The quality of English Language is high, only minor changes are required:

Ln 34: should be 'pioneer'

Ln 37: 'cell types' sounds vague. Maybe: 'various cell types'

Ln 42: should be 'engage'.

Ln 49: should be 'can also'

Ln 53: format the spaces after 'PPPA'

Lns 132 and 133: the phrase 'The molecular identity... ' should be revised. Suggestion: "The molecular identity of mPTP and regulation of its opening by ... have been elucidated'.

Ln 160: 'produce... products' - this part should be revised

Lns 164 and 165 - should be 'NF-kB'

Ln 197 - should be 'influence' 

Ln 245 - should be 'curtails'

Ln 352: should be '... ; it involves...'

Ln 354 - should be 'enables'

Response: We corrected all the typos as advised (and others as well). We also corrected the author name “Xuedong Li” to “Xiaodong Li”.

Reviewer 2 Report

This is a very well written review article on the effect of circadian cycle of liver mitochondria. I enjoyed reading the article. However, I have two suggestions to further improve this review. 

1. If authors can include a section on mtDNA damage, aging and circadian rhythms, that will be a good addition. We all know that mtDNA does not have the protection of Histones unlike nuclear DNA and therefore, more susceptible towards damage which cause aging.

2. Authors can include another section on the effect of Aging induced circadian rhythm disruption on mitochondrial OXPHOS complexes as these are major source of mtROS. 

Author Response

  1. If authors can include a section on mtDNA damage, aging and circadian rhythms, that will be a good addition. We all know that mtDNA does not have the protection of Histones unlike nuclear DNA and therefore, more susceptible towards damage which cause aging.

Response: Thanks for this suggestion. We added new sentences to discuss about mtDNA damage in section 9. “mtDNA is thought to be particularly vulnerable to damages due to its proximity to mitochondrial ROS. However, antioxidant defense, mitochondrial DNA repair, mitochondrial dynamics (fusion and fission) and mitophagy appear sufficient to maintain the integrity of mtDNA at least to the early stage of aging”.  We acknowledged that mtDNA damages may accumulate at later stage of aging.

  1. Authors can include another section on the effect of Aging induced circadian rhythm disruption on mitochondrial OXPHOS complexes as these are major source of mtROS. 

Response: Thanks for this suggestion. We added new sentences to discuss the consequence of  age-changes in subunit composition of OXPHOS complexes in section 9. “Age-related decrease in mtDNA transcripts is also seen in mouse muscle, where nuclear-encoded OXPHOS transcripts are less affected by aging (95). Changes in OXPHOS subunit composition probably lead to inefficient electron transfer and energy production and increased ROS production in various mouse tissues.”

Reviewer 3 Report

The manuscript by Wei Xu and Xuedong Li is a well written, comprehensive review summarizing our current knowledge about an interesting research field. I only have a few suggestions:

1. The authors suggest that in the future it would be important to investigate the effects of time-restricted feeding (tRF)and caloric restriction (CR) on liver mitochondrial function and rhythmicity. I recommend to summarize what we know about the effects of CR and tRF on hepatic clock genes, function and rhythmicity and on mitochondrial (redox) function and rhythmicity in the corresponding chapters. In the current form of the manuscript tRF and CR appear as new terms in the conclusion section and hang in the air a little bit.

2.  I would suggest merging the two figures, as they are similar and if they were placed side by side, it would be easier to interpret the effects of ageing and inflammation on mitochondrial rhythmicity.

English is fine with very few spelling mistakes. 

Author Response

  1. The authors suggest that in the future it would be important to investigate the effects of time-restricted feeding (tRF)and caloric restriction (CR) on liver mitochondrial function and rhythmicity. I recommend to summarize what we know about the effects of CR and tRF on hepatic clock genes, function and rhythmicity and on mitochondrial (redox) function and rhythmicity in the corresponding chapters. In the current form of the manuscript tRF and CR appear as new terms in the conclusion section and hang in the air a little bit.

Response: Thanks for this suggestion. We added new sentences in section 8 about tRF and CR instead of in the “Summary and perspective” section.

  1. I would suggest merging the two figures, as they are similar and if they were placed side by side, it would be easier to interpret the effects of ageing and inflammation on mitochondrial rhythmicity.

Response: We merged the two figures into  one as suggested.

Comments on the Quality of English Language

English is fine with very few spelling mistakes. 

Response: We corrected all the typos as advised.  We also corrected the author name “Xuedong Li” to “Xiaodong Li”.
